

# Untargeted metabolomic analyses of fermented unpolished black rice with melanogenesis inhibition activity

Orrarat Sangkaew[1], Suttida Kaenboot[1], Thumnoon Nhujak[2], Chadin Kulsing[2,3], Nuttanee Tungkijanansin[2], Sittiruk Roytrakul[4] and Chulee Yompakdee[1]

[1] Department of Microbiology, Faculty of Science, Chulalongkorn University, Bangkok, Thailand
[2] Department of Chemistry, Faculty of Science, Chulalongkorn University, Bangkok, Thailand
[3] Metabolomics for Life Sciences Research Unit, Chulalongkorn University, Bangkok, Thailand
[4] National Center for Genetic Engineering and Biotechnology, National Science and Technology Development Agency, Pathum Thani, Thailand

Corresponding author
Chulee Yompakdee,
chulee.y@chula.ac.th

## ABSTRACT

Fermentation of rice can enhance the release of bioactive ingredients and generate diverse microbial metabolites contributing to various functional properties. Previous studies have demonstrated that the mixture of selected microorganisms called "De-E11 starter," comprised of *Rhizopus oryzae*, *Saccharomyces cerevisiae*, *Saccharomycopsis fibuligera* and *Pediococcus pentosaceus* yields fermented unpolished black rice sap (FUBRS) with a melanogenesis inhibition activity. To further understand this fermentation process, we characterized FUBRS and profiled its metabolite composition in comparison to unfermented unpolished black rice (Un-FR), recognizing the substantial enzymatic activity of FUBRS microorganisms and their potential for extensive metabolite production. The results indicated that fermentation decreased the pH, increased total acid content and elevated reducing sugar content. Moreover, significant alterations in phytochemical profiles were observed in FUBRS. In terms of biological activity, fermentation significantly enhanced antioxidant and tyrosinase/melanogenesis inhibitory activities. Untargeted metabolomic analysis utilizing orthogonal projections to latent structures discriminant analysis (OPLS-DA) revealed a clear differentiation in metabolite profiles between FUBRS and Un-FR. Volcano plot analysis (≥2-fold change) indicated a general increase in metabolites, including sugars, phenolic acids, organic acids, and fatty acids, after fermentation. Quantitative analysis confirmed the accumulation of *p*-hydroxybenzoic acid, lactic acid, acetic acid, and succinic acid, that are all known melanogenesis inhibitors. This study provides valuable insights into the characteristics and metabolite profile of FUBRS, and informing strategies for optimizing the fermentation processes to enhance the production of melanogenesis and tyrosinase inhibitory compounds, and identifying key metabolites as critical biomarkers for monitoring and controlling these processes. Together, they will facilitate the efficient and reproducible generation of high-efficacy ingredients for the cosmetic, nutraceutical, and potentially pharmaceutical industries.

# INTRODUCTION

Fermentation has been associated with nutrition-promoting effects for food and is also beneficial for producing or increasing the liberation and bioavailability of phytochemical components in food materials (*Annunziata et al., 2020*). Fermentation is a process in which large molecules are catabolized into other, typically simpler, compounds by microorganisms, leading to marked biochemical changes in the substrate composition. In addition, the microbial strains used in the fermentation process also produce additional, often beneficial, compounds, such as proteins, amino acids, antibiotics, probiotics, and antioxidants, which can result in an increased biological activity and bioavailability (*Sharma et al., 2020*). Hence, the microorganism composition plays a significant if not crucial role in the fermentation process. Besides microorganisms, raw materials also influence the quality of fermented products.

Rice (*Oryza sativa* L) is the pivotal staple food in several countries, especially in Asia. The varieties of rice are usually divided into white rice and pigmented rice. Recently, pigmented rice varieties have received increasing amounts of attention from consumers because of their higher beneficial nutritional and functional attributes than white rice (*Samyor, Das & Deka, 2017*) and, especially, their higher level of phytochemical compounds (*Samyor, Das & Deka, 2017*). Thus, the use of pigmented rice as a raw material for fermentation provides a variety of fermented rice products that are enriched in nutrients with potential benefits for physiological activities.

The fermented rice sap from various pigmented rice cultivars, especially fermented purple plain rice, showed a high free radical scavenging activity, as well as tyrosinase and matrix metalloprotenase-2 (MMP-2) inhibition activities (*Ruksiriwanich et al., 2011*). Likewise, solid-state fermentation of black rice bran with *Aspergillus awamori* and *Aspergillus oryzae* increased the level of total phenolic compounds and the antioxidant and tyrosinase inhibitory activities compared to the unfermented rice (*Shin et al., 2019*). Earlier research showed that the fermented unpolished black rice sap (FUBRS) obtained from unpolished black rice (UBR) and a specific mix of microbes called ''De-E11 starter'' (which includes *Rhizopus oryzae* E1101, *Saccharomyces cerevisiae* E1103, *Saccharomycopsis fibuligera* E1102, and *Pediococcus pentosaceus* E1104), which was isolated from the selected *loogpang* E11, reduced melanin production, while UBR increased melanin levels in B16F10 melanoma cells compared to untreated cells. Moreover, FUBRS was linked to a decreased tyrosinase activity, an essential enzyme in melanogenesis, and the decreased expression level of the melanogenesis related protein; whereas the unfermented rice (Un-FR) did not show any melanogenesis inhibition or tyrosinase inhibitory activity (*Sangkaew & Yompakdee, 2020*). In addition, the metaproteomic and metatranscriptomic analyses revealed that the defined microorganisms in FUBRS produced various enzymes resulting in obtaining many metabolites in FUBRS (*Sangkaew et al., 2020*; *Sangkaew et al., 2023*).

The extensive enzymatic activity of microorganisms producing FUBRS, as demonstrated in earlier studies, underscores their potential to generate a diverse array of metabolites. However, the specific metabolites responsible for FUBRS's biological activity remain largely

uncharacterized, a knowledge gap this study sought to fill through detailed metabolomic analysis.

Metabolomics analysis has emerged as a powerful and increasingly indispensable tool for the comprehensive qualitative and quantitative profiling of metabolite concentrations, encompassing such diverse biochemical classes as amino acids, lipids, and phytochemicals. Leveraging advanced analytical platforms, including $^1$H-NMR, GC-MS, and LC-MS, metabolomics enables the systematic investigation of the dynamic metabolite landscape within biological systems (*Fiehn, 2001*). Especially in the case of fermentation processes, metabolites may also act as nutrients that directly affect the growth of microorganisms and may be related to various health benefits arising from the fermented products (*Tripathi & Giri, 2014*). This study, therefore utilized metabolomics analysis to investigate the characteristic metabolite profile of FUBRS and pinpoint specific compounds correlating with its melanogenesis and/or tyrosinase inhibitory effects. The identified metabolites can serve as critical biomarkers for monitoring and controlling the rice fermentation process, facilitating the efficient and reproducible production of ingredients with high efficacy for applications in the cosmetic, nutraceutical, or potentially pharmaceutical industries.

# MATERIALS AND METHODS

## Materials & chemicals

The microorganisms in the defined De-E11 microbial starter were obtained from the Department of Microbiology, Faculty of Science, Chulalongkorn University. All media for cell growth, such as Dulbecco's modified Eagle's medium (DMEM) high glucose, fetal bovine serum (FBS), and trypsin-EDTA (0.25%) were purchased from Gibco-BRL Inc., USA. The high-performance liquid chromatography (HPLC)-grade chemicals were purchased from Merck, Germany. Folin-Ciocalteu reagent, 2,2-diphenyl-l-picrylhydrazyl (DPPH), 2,2-azino-bis-(3-ethylbenzothiazoline-6-sulfonic acid) diammonium salt (ABTS), and mushroom tyrosinase enzyme were purchased from Sigma-Aldrich, USA. Kojic acid and l-3,4-dihydroxyphenylalanine (L-DOPA) were purchased from Tokyo Chemical Industry Co., Ltd., Japan. The UBR was purchased from L.H. Rice International Co., Ltd. (Nakhon Pathom, Thailand).

## Sample preparations

The sample preparation was performed as previously described (*Sangkaew & Yompakdee, 2020*). Briefly, 20 g of UBR was added to 40 mL of distilled water and autoclaved at 121 °C for 15 min. After being cooled, the defined De-E11 microbial starter, comprised of *R. oryzae* E1101, *S. cerevisiae* E1103, *S. fibuligera* E1102, and *P. pentosaceus* E1104 at $1 \times 10^4$, $2 \times 10^4$, $1 \times 10^3$, and $3 \times 10^8$ colony forming units (CFU)/g, respectively, was mixed with cooked UBR and incubated at 30 °C for 12 days in a closed sterilized bottle. The fermented liquid was then collected (called FUBRS). For the Un-FR, 20 g of UBR was mixed with 40 mL of distilled water and boiled at 80 °C for 15 min, then centrifuged as above and the supernatant (Un-FR) was harvested. Both the Un-FR and FUBRS samples were then evaporated, resuspended in 20 mL of distilled water, and kept at −20 °C for further study.

## Physicochemical analysis

The pH of the samples was measured using a pH meter (S-20K; Mettler-Toledo, Greifensee, Switzerland). The titratable acidity was measured by titrating the samples with 0.1 N NaOH using phenolphthalein as the color indicator (*Woldemariam et al., 2014*). The total reducing sugars was measured using the 3,5-dinitro salicylic acid (DNS) method and expressed as mg glucose equivalents per mL sample (mg GE/mL of sample) (*Zeng et al., 2017*).

## Determination of total phytochemical contents
### Total phenolic content

The total phenolic content (TPC) was measured by the Folin–Ciocalteu method as adapted from *Tawaha et al. (2007)*. Briefly, 18 µL of sample and 36 µL of 10% (v/v) Folin-Ciocaltue reagent were mixed and incubated at room temperature for 5 min. Next, 146 µL of 350 mM $Na_2CO_3$ was added, incubated for 30 min, and then the absorbance was measured at 765 nm ($A_{765}$). The results are expressed as mg of gallic acid per mL (mg GAE/mL) of sample.

### Total flavonoid content

The total flavonoid content (TFC) was measured by the aluminium chloride ($AlCl_3$) colorimetric method, as adapted from *Tian et al. (2011)*. Briefly, 12.5 µL of sample and 7.5 µL of 5% (w/v) $NaNO_3$ were mixed and incubated at room temperature for 5 min. Next, 15 µL of 10% (w/v) $AlCl_3$ was added and incubated for 6 min. Then, 75 µL of 1 M NaOH was added and the mixture was adjusted with distilled water to 200 µL before the absorbance was measured at 510 nm ($A_{510}$). The results are expressed as mg of quercetin per mL (mg QuE/mL) of sample.

### Total anthocyanins content

The total anthocyanins content (TAC) was determined using the pH differential method described by *Zheng et al. (2022)*. The sample was separately diluted with 0.025 M KCl-HCl buffer (pH 1.0) and 0.4 M sodium acetate buffer (pH 4.5), incubated for 30 min at room temperature, and then the absorbance values at 512 nm ($A_{512}$) and 700 nm ($A_{700}$) of the two different pH dilutions were measured. The results are expressed as mg of cyanidin-3-glucoside equivalents per mL (mg CGE/mL) of sample, following the method of *Zheng et al. (2022)*.

## Biological activity analysis
### The DPPH radical scavenging activity

The DPPH scavenging ability assay was used to evaluate the antioxidant activity of Un-FR and FUBRS using the method of *Tachibana et al. (2001)* with some modifications. The samples were incubated in 25 mg/L DPPH solution for 30 min and then the DPPH radical scavenging activity was measured by spectrophotometry at 515 nm ($OD_{515}$). Ascorbic acid (10 µg/mL) was used as the positive control. The antioxidant activity was calculated from Eq. (1):

$$\text{DPPH scavenging activity (\%)} = [(A - B)/A] \times 100, \tag{1}$$

where A and B are the $OD_{515}$ values of the without and with test sample, respectively.

### ABTS radical scavenging assay

The antioxidant activity was determined as the ABTS radical scavenging activity. For the assay, ABTS$^+$ radicals were prepared by mixing an ABTS stock solution (seven mM in water) with 2.45 mM potassium persulfate. This mixture was allowed to stand for 12–16 h at room temperature in the dark until reaching a stable oxidative state. The ABTS$^+$ solution was diluted with 20 mM sodium acetate buffer (pH 4.5) to an absorbance of $0.70 \pm 0.01$ at 734 nm (A$_{734}$). The reaction was started by the addition of 40 µl of sample to 160 µl of the diluted ABTS$^+$ solution, incubated for 10 min at room temperature to allow ABTS$^+$ bleaching, and then the A$_{734}$ was measured by spectrophotometry. The antioxidant activity was then calculated from Eq. (2):

$$\text{ABTS scavenging activity (\%)} = [(A - B)/A] \times 100, \tag{2}$$

where A and B are the A$_{734}$ values of the without and with test sample, respectively.

### Anti-mushroom tyrosinase activity (in vitro)

The samples of Un-FR and FUBRS were examined for tyrosinase inhibiting activity by evaluating the mushroom tyrosinase activity *via* the reported dopachrome method (*Zengin et al., 2015*) except with slight modification. The reaction mixture, which consisted of 80 µL of 50 mM potassium phosphate buffer (pH 6.8), 40 µL of the sample, and 40 µL of mushroom tyrosinase (50 U/mL), was mixed and then incubated at room temperature for 10 min. Next, 40 µL of 1.5 mM L-DOPA was added and incubated at room temperature for 10 min. The amount of dopachrome in the reaction mixture was then measured by spectrophotometer at 492 nm (A$_{492}$). Kojic acid (0.03 mg/mL) was used as the positive control. The percent inhibition of tyrosinase activity was calculated from Eq. (3):

$$\text{Mushroom tyrosinase inhibition (\%)} = [(A - B)/A] \times 100 \tag{3}$$

where A and B are the A$_{492}$ values of the without and with test sample, respectively.

### Melanogenesis inhibition activity

Melanogenesis inhibition activity was determined by measurement of the melanin content in the B16F10 melanoma cell line as previously described (*Sangkaew & Yompakdee, 2020*). The B16F10 cell line (ATCCCCL-6475$^{™}$) was cultured in DMEM supplemented with 10% (v/v) FBS, penicillin (100 U/mL), and streptomycin (100 mg/mL) in six-well plates ($5 \times 10^4$ cells/well) for 24 h at 37 °C under a humidified 95:5 (v/v) air: CO$_2$ atmosphere. The cells were then treated with Un-FR or FUBRS at the indicated final concentration and incubated for 72 h. After incubation, the cells were harvested and solubilized in 1 N NaOH at 60 °C for 60 min, and then the absorbance of the cell suspension was measured by spectrophotometer at 405 nm (A$_{405}$). The relative melanogenesis inhibition (%) was calculated from Eq. (4):

$$\text{Relative melanogenesis inhibition (\%)} = [1 - (A \div B)/(C \div D)] \times 100 \tag{4}$$

where A and C are the A$_{405}$ values of the treated cells and untreated cells, respectively, and B and D are the protein concentrations (mg/mL) of the treated cells and untreated cells, respectively.
## Untargeted metabolites profiling by gas chromatography-mass spectrometry (GC-MS)

### Extraction and derivatization of metabolites

Non-targeted analysis was adopted from *Lee et al. (2012)*. Either one mL of Un-FR and/or one mL of FUBRS were shaken in 10 mL of a 2.5:1:1 (v/v/v) mixture of methanol: water: chloroform at room temperature for 24 h and then centrifuged at 5,000×g for 8 min at 20 °C. The supernatant was collected, evaporated to dryness, and reconstituted in one mL of pyridine. Next, 50 µL of this extract was derivatized with 100 µL of *N, O*-bis(trimethylsilyl)trifluoroacetamide (BSTFA) containing 1% (v/v) trimethylchlorosilane (TMCS) and heated at 60 °C for 1 h prior to GC-MS injection.

For fatty acid analysis, each UBR or FUBRS sample (one mL) was shaken in 25 mL of a 2:1 (v/v) dichloromethane: methanol mixture and then boiled at 100 °C for 5 min. The sample was filtered and evaporated to dryness before derivatization of its fatty acids. After extraction of the fatty acids, the lipid extracts were methylated and converted into fatty acid methyl esters (FAMEs) using a method adopted from *Li-Beisson et al. (2013)*. Briefly, one mL of NaOH (0.5 M in methanol) was added and heated at 100 °C for 15 min. After being cooled, 20% (v/v) boron trifluoride ($BF_3$) solution in methanol was added to the sample and heated at 100 °C for 1 min. Then, 500 µL of hexane and five mL of saturated NaCl solution was added, mixed, and the sample was centrifuged at 1,000×g for 10 min. The organic layer was harvested and kept in a vial for subsequent GC-MS analysis.

### GC-MS analysis

Chromatographic separation was performed on a GC-MS system (Agilent Technologies) using a HP-5 ms (30 m × 250 µm × 0.25 µm; Agilent Technologies) capillary column. The oven and injector temperatures were set at 80 °C. Helium was used as the column carrier gas at a constant flow rate of 0.8 mL/min. For fatty acid analysis, the temperature and constant flow rate were set at 240 °C and 2.25 mL/min, respectively. Identification of the untargeted compounds was performed by comparing the MS fragmentation patterns with those of reference compounds, including the mass spectra in the NIST database.

## Quantitative determination of metabolites in the FUBRS sample

The FUBRS sample were extracted by solid-phase extraction using Sep-Pak C18 column cartridges (Sep-Pak Waters, Milford, MA, USA) according to *Flores et al. (2012)* with some modifications. Briefly, the cartridge was first conditioned with one mL of methanol and then with 1.0 mL of milliQ water. Next, the sample was loaded and washed with 50 mL of milliQ water. The analyte was eluted with five mL of 0.1% (w/v) HCl in methanol, evaporated to dryness in a rotary evaporator at 40 °C, re-dissolved in 50% (v/v) methanol, and then kept at −20 °C for further analysis.

The metabolites in the sample extract were determined by high-performance liquid chromatography (HPLC). Initially, 1.0 mL of flow through from FUBRS was performed using an Agilent HPLC 1260 Infinity II HPLC system (Agilent Technologies) with a UV/Vis detector (SPD-20A) monitoring at a wavelength of 210 nm, and a 5 µm, 4.6 × 150 mm Mightisil RP18 column (Waters). The sample injection volume was five µL and elution was performed in an isocratic mode with 0.02 M $Na_2HPO_4$, pH 2 at a flow rate of 0.1 mL/min.

**Table 1  Physicochemical properties of the Un-FR and FUBRS.**

| Chemical properties | Un-FR | FUBRS |
|---|---|---|
| pH | 5.98 ± 0.09 | 3.38 ± 0.01[***] |
| % Titratable acidity | 0.12 ± 0.02 | 11.18 ± 0.03[***] |
| Reducing sugar (mg/mL) | 2.00 ± 0.14 | 15.99 ± 0.21[***] |

**Notes.**
Values are presented as mean ± standard error ($n = 3$).
[***] Statistically significant ($p < 0.001$) differences between FUBRS and Un-FR.

## Statistical analysis

All experiments were performed in triplicate, and the data in the tables are presented as the respective mean ± one standard error. The statistical analysis of data was performed using GraphPad Prism 9.0.2 (San Diego, CA, USA), and differences among groups were determined by one-way analysis of variance (ANOVA) with Dunnett's Multiple Comparisons test. To elucidate the differential metabolites, the metabolites were imported into the MetaboAnalyst 6.0 (https://www.metaboanalyst.ca/) online software.

## RESULTS AND DISCUSSIONS

### Physical characteristics and phytochemical content of FUBRS

Fermentation of UBR by the microorganisms in the De-E11 is known to effect the final product (*Sangkaew et al., 2020*; *Sangkaew et al., 2023*). The physicochemical properties and phytochemicals are important indicators that can be used to evaluate the degree of fermentation. Their characterization in the FUBRS and Un-FR samples revealed that the total acid contents increased markedly (93-fold) while the pH decreased in FURBS compared to in the Un-FR (Table 1). Moreover, the level of reducing-sugars was also markedly increased (8-fold) in the FUBRS compared to that in the Un-FR. In the fermentation process, starch is digested into the reducing sugar glucose, an important substrate for alcohol fermentation that greatly impacts the acidity, taste, and alcohol content (*Kim et al., 2007*).

The total phytochemical contents, as the TPC, TFC, and TAC, of the Un-FR and FUBRS are presented in Table 2. The results revealed that the Un-FR had a significantly higher phytochemical content than FUBRS. A similar result was reported in rice wine fermentation, and was suggested that the reduction in the TPC upon fermentation might be related to the utilization of phenolic compounds by microorganisms (*Mu et al., 2019*). Interestingly, the color of the FUBRS sample was lighter than that of the Un-FR sample (Fig. S1), which is consistent with the higher (over 3.5-fold) TAC in the Un-FR than in the FUBRS. Transformation of anthocyanins is mediated by its reaction with yeast metabolites, such as pyruvic acid and acetaldehyde (*De Freitas & Mateus, 2011*). Moreover, anthocyanins are also degraded into their aglycone forms by lactic acid bacteria during the fermentation process (*Braga et al., 2018*). Hence, these results suggested that the fermentation of UBR with the defined De-E11 microbial starter may convert the compounds in UBR into other compounds, which leads to the reduced phytochemical content in FUBRS.

**Table 2  Phytochemical contents of Un-FR and FUBRS.**

| Phytochemical contents | Un-FR | FUBRS |
|---|---|---|
| Total phenolic content (TPC) ($\mu$g GAE/mL of sample) | 1,335.0 $\pm$ 2.0 | 459.0 $\pm$ 1.0[***] |
| Total flavonoid content (TFC) ($\mu$g QuE/mL of sample) | 40.5 $\pm$ 0.8 | 20.3 $\pm$ 1.4[***] |
| Total anthocyanins content (TAC) ($\mu$g CGE/mL of sample) | 14,612.0 $\pm$ 460.0 | 4,063.0 $\pm$ 790.0[***] |

Notes.

Values are presented as mean $\pm$ standard error ($n = 3$).

[***]Statistically significant ($p < 0.001$) differences between FUBRS and Un-FR.

## Biological activity of FUBRS

Fermented rice, especially colored rice, is rich in various bioactive components and has many health benefits (*Ruksiriwanich et al., 2011*). Hence, we investigated the biological activities, in terms of the antioxidant activity, mushroom tyrosinase activity, and melanogenesis inhibition activity, of the FUBRS and Un-FR samples.

### Antioxidant characteristics of the FUBRS and Un-FR samples

The DPPH and ABTS assays are the most popular and convenient methods to determine the antioxidant activity. Specifically, the ABTS assay is based on the generation of blue/green ABTS$^+$ ions that can be reduced by antioxidants, whereas the DPPH assay is based on the reduction of the purple DPPH to 1,1-diphenyl-2-picryl hydrazine (*Floegel et al., 2011*; *Sánchez et al., 2007*). Hence, the antioxidant capacity of the FUBRS and Un-FR samples was determined by both methods using ascorbic acid as the positive control. As depicted in Fig. 1, both the FUBRS and Un-FR samples possessed antioxidant activities with a clear concentration-dependent radical scavenging activity in both the DPPH and ABTS assays. However, Un-FR had a higher antioxidant activity than FUBRS in both assays. These results are in accord with a previous report that fermentation of black rice bran decreased the antioxidant activity, TPC, and TAC (*Yoon et al., 2015*). Moreover, our results revealed that the antioxidant activity of FUBRS and Un-FR (Fig. 1) was consistent with the TPC (Table 2). Thus, the reduction in the antioxidant activity after fermentation may be associated with the degradation of high antioxidant activity phenolic compounds into phenolic or other compounds of a lower or no antioxidant activity (*Gan et al., 2017*; *Melini & Melini, 2021*).

### Mushroom tyrosinase inhibition activity

When skin is exposed to UV rays it causes an increase in tyrosinase expression, a key enzyme in melanin biosynthesis (*Chan et al., 2014*). Mushroom tyrosinase has been extensively utilized as a model system for the screening of tyrosinase inhibitors and in melanogenic studies (*Zolghadri et al., 2019*). Hence, the potential inhibitory effect of FUBRS and Un-FR on mushroom tyrosinase activity was evaluated using Kojic acid as the positive standard. The results (Fig. 2) clearly revealed that the tyrosinase inhibition activity was significantly increased following fermentation (FUBRS *vs.* Un-FR). In accord, several researchers have reported that fermentation could enhance the tyrosinase inhibition activity of raw materials (*Abd Razak et al., 2017*; *Chan et al., 2014*; *Chen et al., 2013*; *Sangkaew & Yompakdee, 2023*).

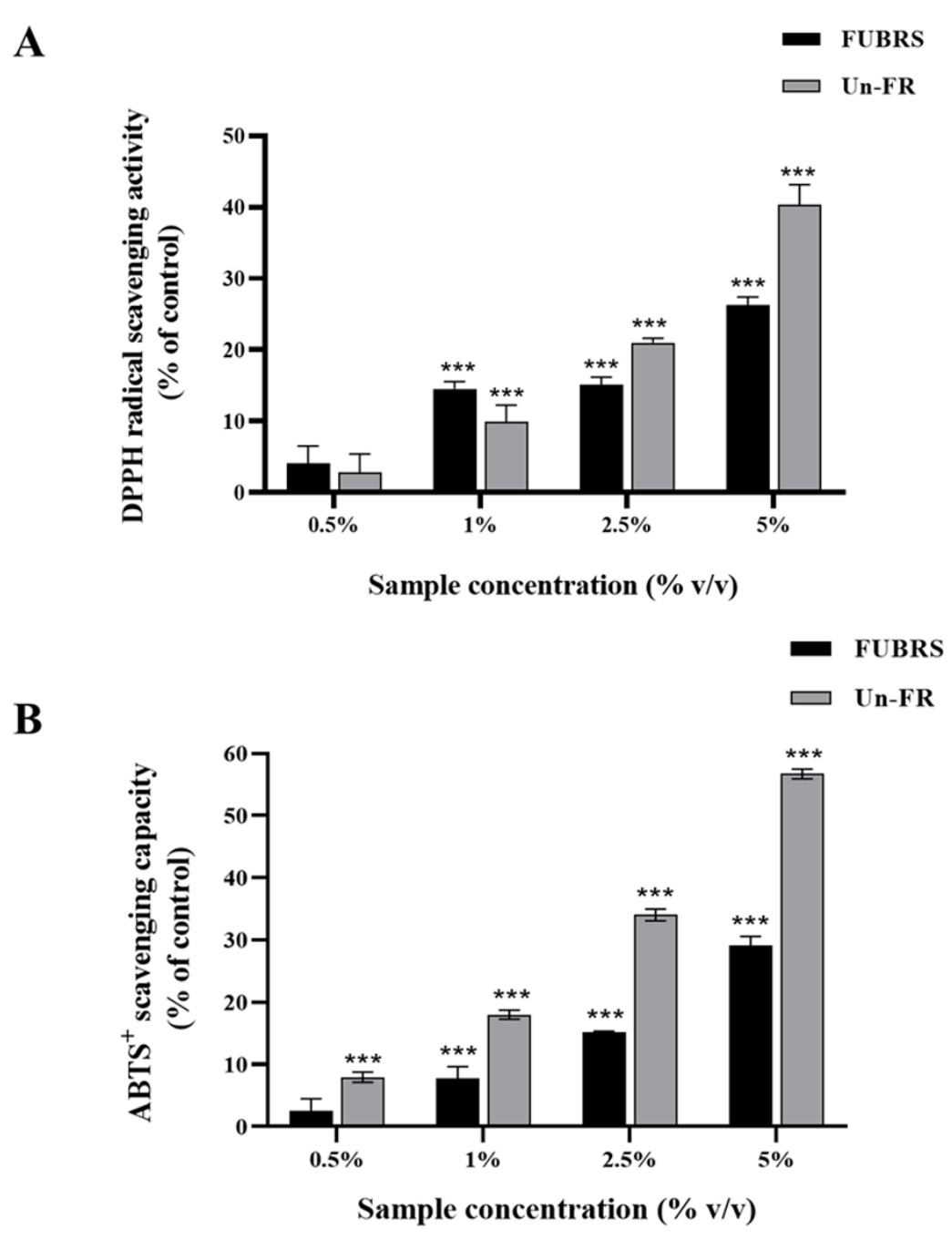

**Figure 1 Antioxidant characteristics of the FUBRS and Un-FR samples.** Both samples were determined using the (A) DPPH scavenging capacity and (B) ABTS + radical scavenging capacity assays. Results are presented as percentages of the control (without samples), and the data are presented as the respective mean ± SD from three independent experiments. ***Significant ($p < 0.001$).

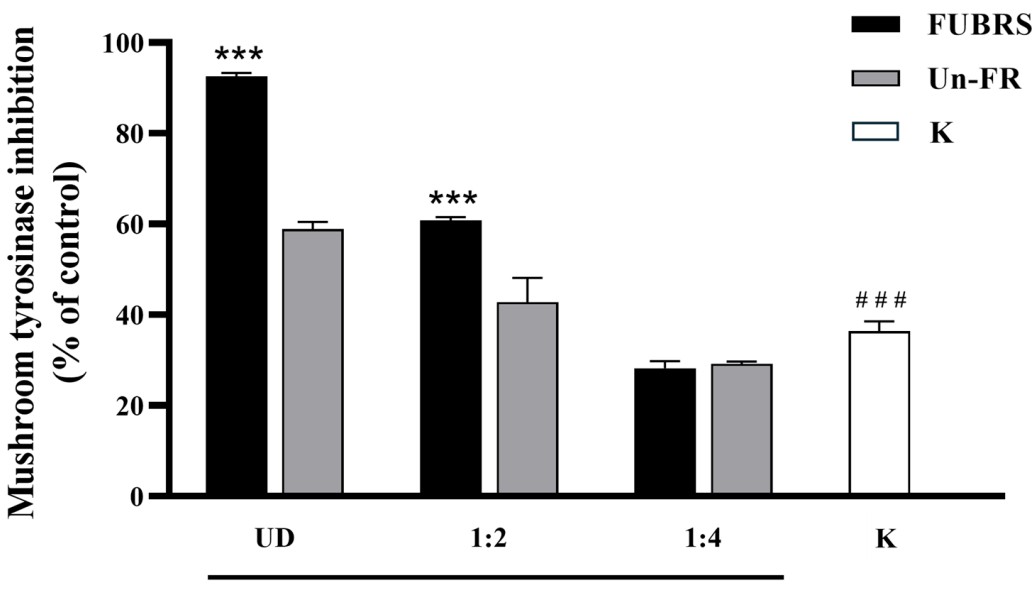

**Figure 2 Mushroom tyrosinase inhibition activity of the FUBRS and Un-FR samples.** The inhibitory activity of various concentrations of FUBRS and Un-FR against mushroom tyrosinase was determined. Kojic acid (K, 1 mM) served as a positive control. UD indicates undiluted extract. The results are presented as percentages of the control (without samples), and the data are presented as the mean $\pm$ SD from three independent experiments. ***Significantly ($p < 0.001$) different compared with the Un-FR. # # # Significantly ($p < 0.001$) different compared with the control.

### *Melanogenesis inhibition activity*

To determine the melanogenesis inhibition activity, 5% (v/v) Un-FR and FUBRS at three different concentrations (1%, 2.5% and 5% (v/v)) were evaluated with the B16F10 melanoma cell line. The results showed that FUBRS significantly inhibited melanin production in B16F10 melanoma cells in a dose-dependent manner, whereas Un-FR showed no such activity (Fig. 3). Interestingly, Un-FR and FUBRS both exhibited anti-mushroom tyrosinase activity (Fig. 2), but Un-FR showed no significant effect on the reduction of melanin content in B16F10 cells even though it was at a higher dose of 5% (v/v) than the FUBRS (1% (v/v)) (Fig. 3). Although mushroom tyrosinase is used for anti-tyrosinase activity screening, it does not appear to be sufficiently sensitive to measure the potential of the active substances, such as FUBRS, as a whitening agent. There are significant differences between mushroom tyrosinase and mammalian tyrosinase in terms of their structural, molecular, and kinetic properties, as well as their localization. Therefore, a more appropriate method might be to use the melanogenesis inhibition results from the cell-based assay (*Promden et al., 2018*; *Song et al., 2009*).

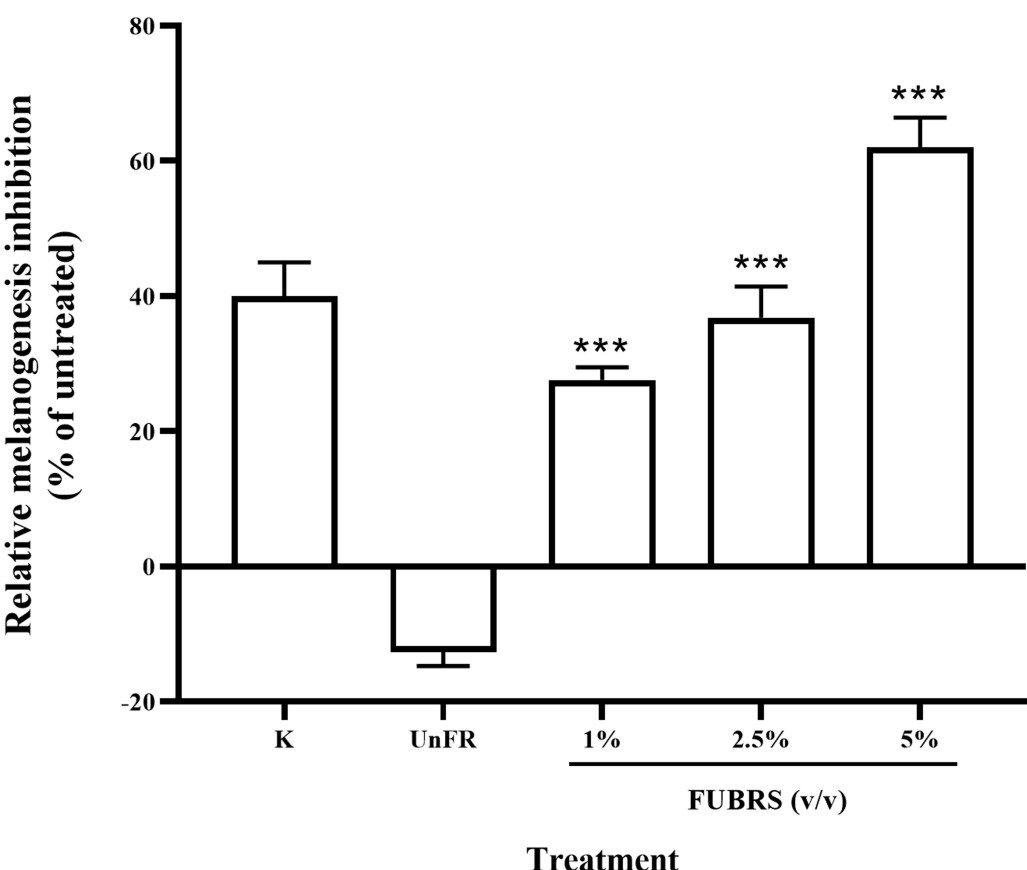

**Figure 3 Melanogenesis inhibition activity of FUBRS and Un-FR samples.** B16F10 melanoma cells were treated with one mM kojic acid (K) as a positive control, Un-FR at 5% (v/v) and different concentrations (1%, 2.5%, and 5% (v/v)) of FUBRS. The results are expressed as a percentage of the untreated group, and the data are shown as the respective mean ± SD from three independent experiments. ***Significantly ($p < 0.001$) different values compared with the control.

## Untargeted metabolic analysis of Un-FR and FUBRS
### Multivariate analysis of Un-FR and FUBRS

Metabolomics based on GC-MS has been successfully applied in the study of dynamic changes in the metabolite profiles in rice koji and rice wine (*Lee da et al., 2016*; *Mu et al., 2019*). Therefore, the metabolites in Un-FR and FUBRS were investigated using GC-MS. To initialize the systemic alteration of metabolites, multivariate analysis by orthogonal projections to latent structures discriminant analysis (OPLS-DA), a common classification method, was performed using the online MetaboAnalyst 6.0 software to identify the difference between the two groups.

The OPLS-DA score plot of the GC-MS data from the Un-FR and FUBRS samples showed a clear separation (Fig. 4), implying that the metabolic profile was significantly altered after fermentation. Moreover, the R2X, R2Y, and Q2 values, which are the critical parameters used to evaluate the OPLS-DA model, were 0.772, 0.935 and 0.886, respectively, in the metabolites group (Fig. S2A) and 0.556, 0.868 and 0.794, respectively, in the fatty acid

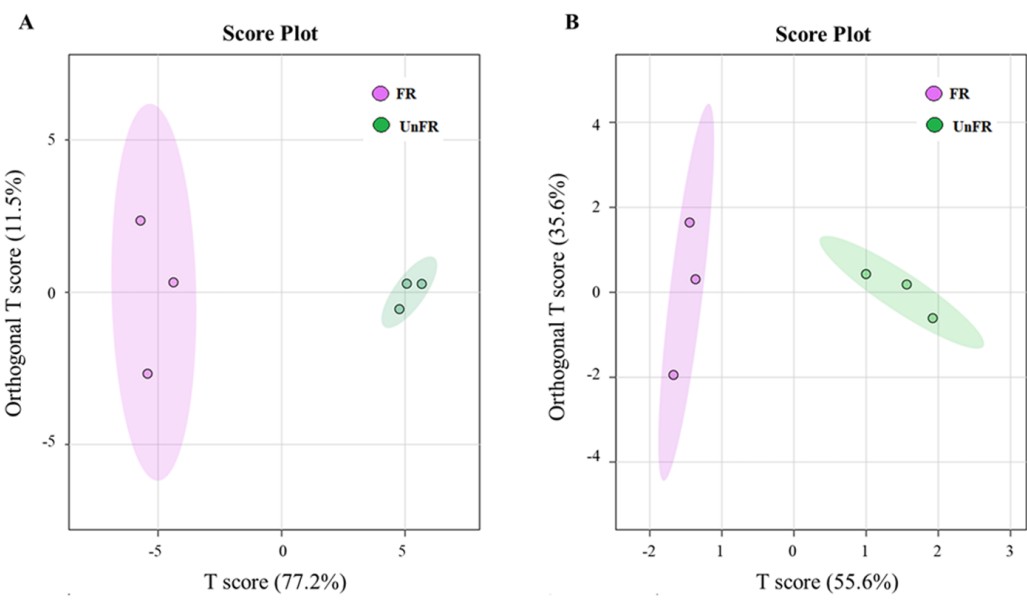

**Figure 4  Orthogonal projections to latent structures discriminant analysis (OPLS-DA) scores of compounds in FUBRS and Un-FR.** Differences in the (A) metabolites and (B) fatty acids between FUBRS and Un-FR were investigated using GC-MS and analyzed by OPLS-DA.

group (Fig. S2B). Note that R2 and Q2 represent the explanatory ratio and predictability, respectively, where the closer the value is to 1, the higher the model accuracy. Generally, when R2 and Q2 > 0.5, the model fits well.

### *Identification of significantly different metabolites in Un-FR and FUBRS*

To investigate the changes in metabolite abundance between Un-FR and FUBRS, a heatmap was constructed using the MetaboAnalyst 6.0 software. In total, 42 metabolites were identified, comprised of 17 sugars and sugar alcohols, nine amino acids, 13 organic acids, and three phenolic acids (Fig. 5). The result indicated that most of these metabolites were increased after fermentation. In the fatty acids group, five fatty acids (stearic, oleic, linoleic, palmitic, and myristic acids) were detected (Fig. 6), of which oleic and myristic acids were increased while linoleic acid was decreased in FUBRS compared to Un-FR.

Furthermore, change in the abundance by more than a two-fold change of 47 metabolite (17 sugars and sugar alcohols, nine amino acids, 13 organic acids, three phenolic acids and five fatty acids) between both samples was analyzed, with the results shown in the volcano plot. The volcano plot illustrated that among all these metabolites, 25 were significantly increased and nine were decreased in FUBRS compared to Un-FR, whereas 13 metabolites were not significantly different (Fig. 7 and Table S1).

During the fermentation process, rice starch is hydrolyzed and then saccharified to sugars and then sugar alcohols by enzymes from the microorganisms. Sugars are generally consumed as carbon sources, providing energy for proliferation and growth of the microorganisms *via* carbohydrate metabolic pathways that led to the production of other metabolites, such as amino acids, organic acids, and fatty acids (*Zhu et al., 2004*).

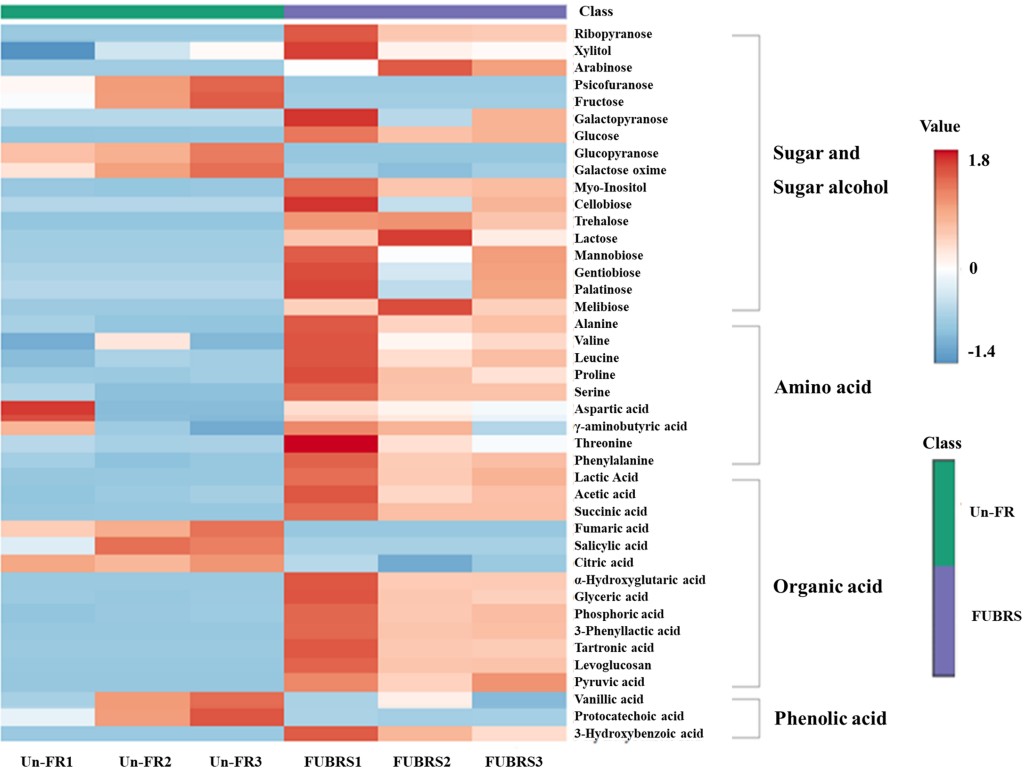

**Figure 5** **Heatmap of the hierarchical clustering of metabolites in Un-FR and FUBRS.** The metabolites in Un-FR and FUBRS were analyzed using the MetaboAnalyst 6.0 software analysis of the GC/MS data. The horizontal and vertical coordinates represent the sample name and the different metabolites, respectively. From blue to red indicates metabolite expression abundance from low to high.

The level of detected sugars and sugar alcohols, namely: ribopyranose, arabinose, glucose, myo-inositol, trehalose, lactose, mannobiose, and melibiose in FUBRS was higher than those in Un-FR. Five amino acids (alanine, leucine, proline, serine, and phenylalanine) were significantly increased after the fermentation process.

Generally, the acidity of a sample results from organic acids, which can dramatically affect the organoleptic quality of the fermented product and can inhibit the growth of microbes. They mainly come from microbial metabolism (*Mato, Suárez-Luque & Huidobro, 2005*). Among the detected 13 organic acids, 10 (lactic, acetic, succinic, α-hydroxy glutaric, glyceric, phosphoric, 3-phenyllactic, tartronic and pyruvic acids as well as levoglucosan) were increased in the fermentation process (FURBS *vs.* Un-FR).

Phenolic compounds in the fermented product are mainly derived from plant materials that are associated with the quality of product (*Martins et al., 2011*). The level of 3-hydroxybenzoic acid was higher in the FUBRS than in the Un-FR samples. Moreover, the fatty acid (oleic acid) content was also increased after fermentation (Fig. 7). Table S2 illustrates the metabolites found in FUBRS which have previously been reported to have an inhibitory activity of melanogenesis or tyrosinase, where the most active metabolites were increased after fermentation.

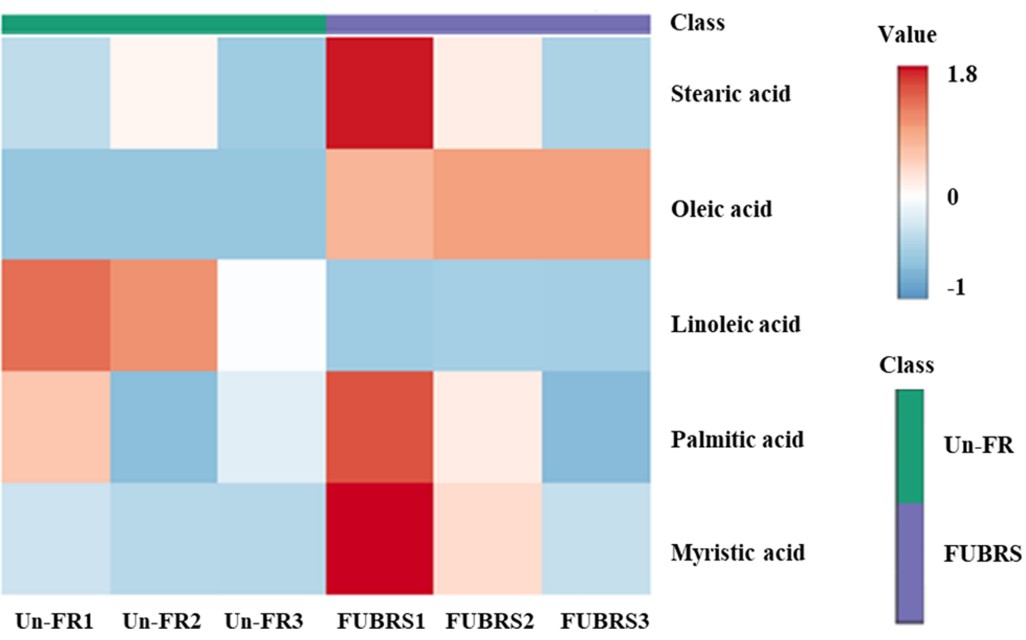

**Figure 6** **Heatmap of the hierarchical clustering of fatty acids in Un-FR and FUBRS.** The fatty acids in Un-FR and FUBRS were analyzed using the MetaboAnalyst 6.0 software analysis of the GC/MS data. The horizontal and vertical coordinates represent the sample name and the different metabolites, respectively. From blue to red indicates metabolite expression abundance from low to high.

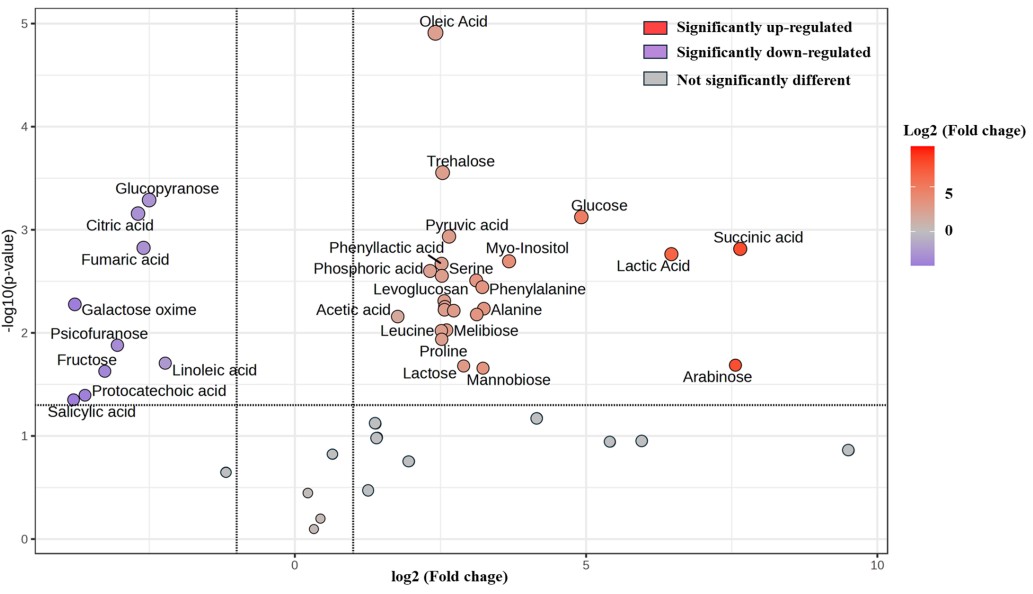

**Figure 7** **Volcano plot of the differential metabolite abundance in FUBRS compared to Un-FR.** Metabolites exhibiting a ≥ 2-fold change and a *p*-value < 0.05 are highlighted. Red and purple circles indicate the significantly up-regulated and down-regulated metabolites in FUBRS, respectively. Gray circles indicate no significantly different metabolite abundance between FUBRS and Un-FR.

Table 3   The contents of some metabolites in the FUBRS.

| Compounds | Amount (ng/mL) |
| --- | --- |
| *p*-Hydroxybenzoic acid | $193.0 \pm 1.0$ |
| Lactic acid | $947.0 \pm 10.0$ |
| Acetic acid | $420.0 \pm 13.0$ |
| Succinic acid | $41.7 \pm 7.60$ |

Notes.

Data are shown as the respective mean $\pm$ SD from three independent experiments, each performed in triplicate.

Our results are consistent with previous studies, which reported that the microorganisms in De-E11 starter expressed the genes or proteins involved in the synthesis of many melanogenesis inhibitors, such as amino acids, organic acids, sugars, phenolic acids, and fatty acids (*Sangkaew et al., 2020*; *Sangkaew et al., 2023*).

## Quantitative determination of some selected metabolites in the FUBRS sample

To determine the quantitative level of metabolites in the FUBRS sample, 1-phenolic acid (*p*-hydroxybenzoic acid) and three organic acids (lactic, acetic, and succinic acids), from the untargeted metabolite results (Figs. 5 and 7) that were significantly increased after fermentation (Fig. 8) were used as representative compounds for quantitative determination in the FUBRS sample (Table 3). These metabolites were selected because they have been reported to have a tyrosinase or melanogenesis inhibition activity (*Chan et al., 2014*; *Yamamoto et al., 2006*). Among the identified organic acids, lactic acid showed the highest level at $947 \pm 10$ ng/mL, followed by acetic acid ($420 \pm 13$ ng/mL) and succinic acid ($41.7 \pm 7.6$ ng/mL), respectively, while *p*-hydroxybenzoic acid was at $193 \pm 1$ ng/mL.

## CONCLUSIONS

The characteristics and metabolites associated with tyrosinase or melanogenesis inhibition activities in FUBRS that were produced from fermenting UBR with the De-E11 microbial starter were investigated in this study. Regarding the physicochemical property, the biological activity, and the metabolite profile, the FUBRS and Un-FR were significantly different. Compared with Un-FR, the FUBRS showed an improved mushroom tyrosinase and melanogenesis inhibitory activity; however, it had slightly decreased antioxidant properties. In addition, the metabolite profiling, including amino acids, organic acids, sugars, phenolic acids, and fatty acids detected by GC-MS analysis, indicated that most of the compounds were significantly increased after fermentation. Hence, these results suggested that the De-E11 starter may increase or convert the compounds in UBR into the other compounds. Additionally, quantitative analysis of some metabolites known to have tyrosinase or melanogenesis inhibitory activity (*p*-hydroxybenzoic, lactic, acetic, and succinic acids) revealed that they were significantly increased after fermentation (FUBRS *vs*. Un-FR).

The metabolic events during rice fermentation are summarized schematically in Fig. 9, beginning with the degradation of rice grain components into sugars and the release of

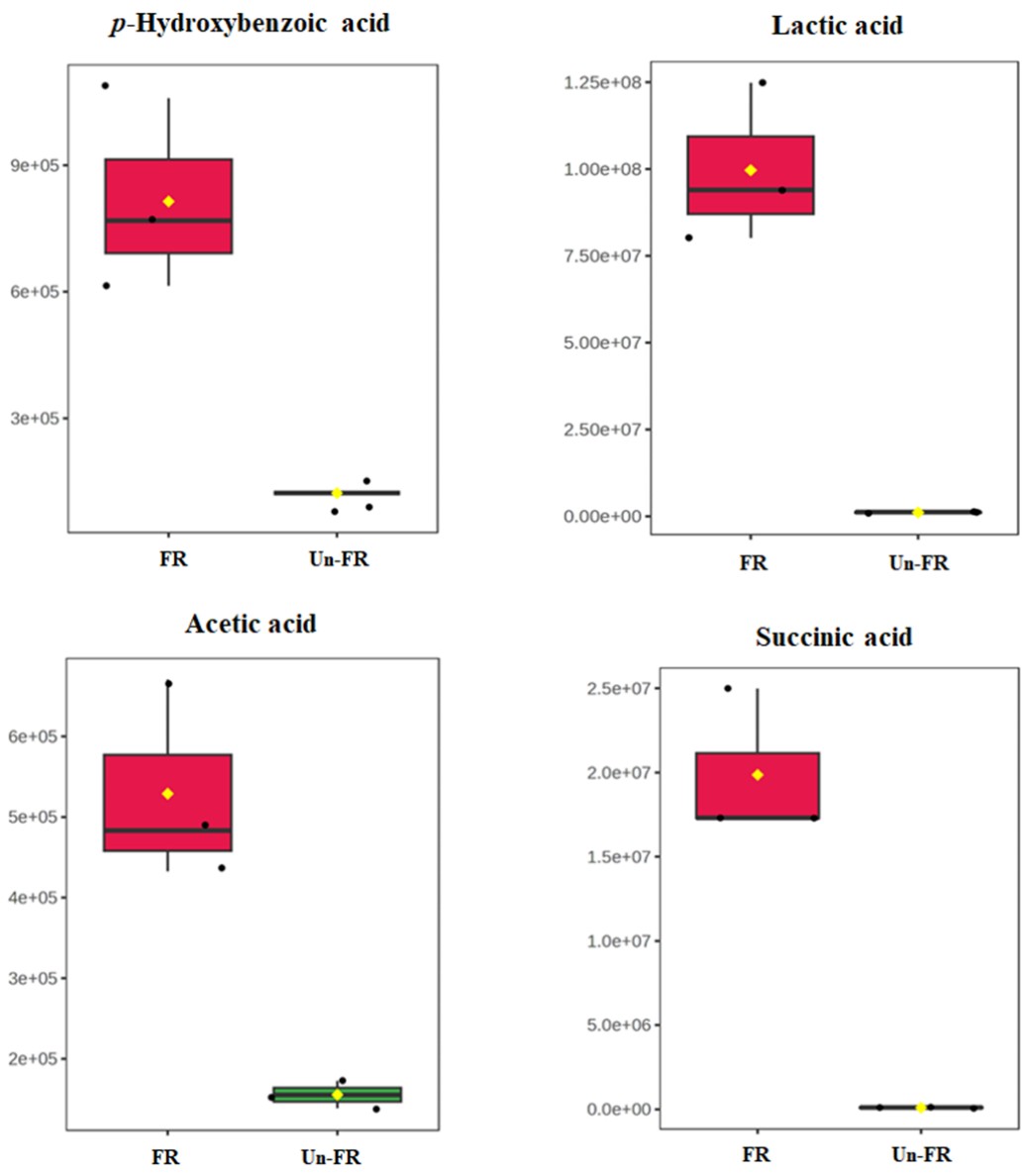

**Figure 8** **Boxplots of selected metabolites with significantly different concentrations in FUBRS and Un-FR.** The concentration levels of *p*-hydroxybenzoic acid, lactic acid, acetic acid and succinic acid that were significantly different (*p*-value < 0.1) between FUBRS (red) and Un-FR (green) were plotted using the MetaboAnalyst 6.0 software analysis.

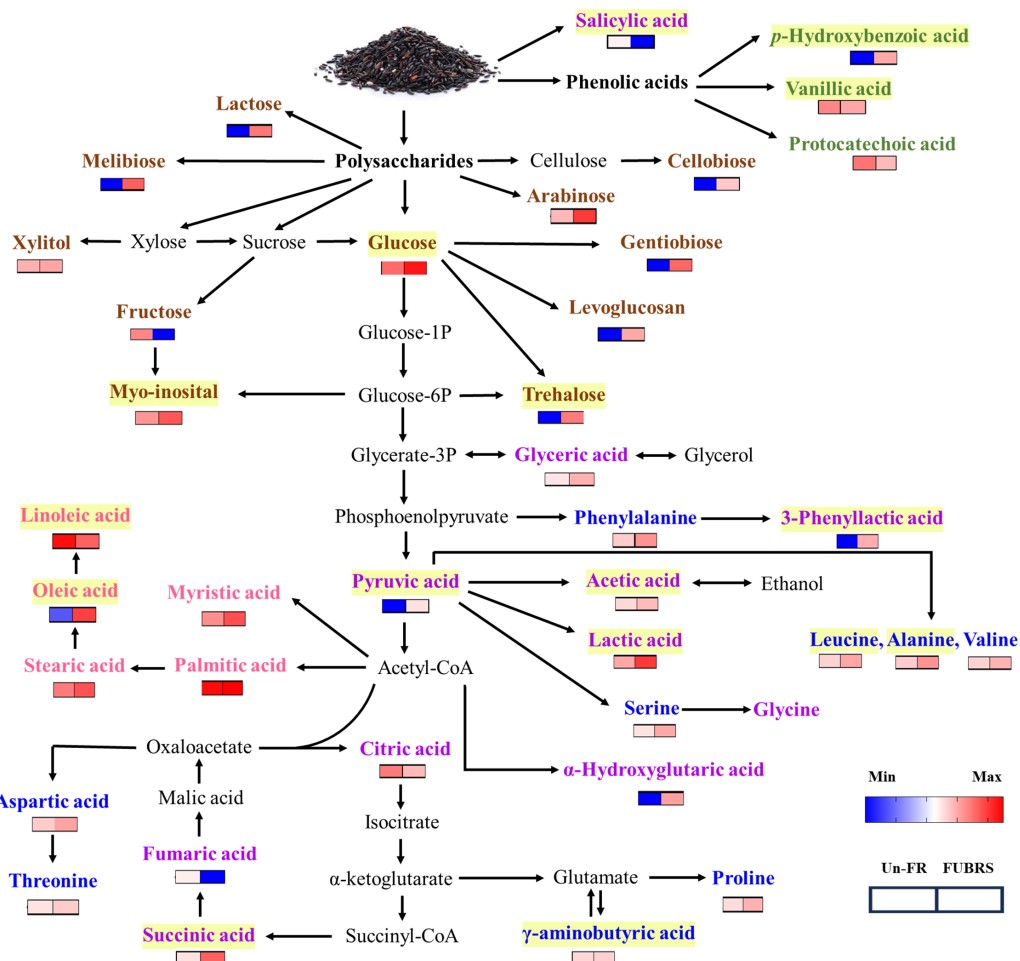

**Figure 9** Overview of the metabolites involved in the metabolic pathway during UBR fermentation to FUBRS. The relative content of all the metabolite groups except the phenolic group was obtained from GC/MS analysis. The relative contents were log$_{10}$-transformed average values and missing values were treated as zeros. The data were then visualized as a heatmap using GraphPad Prism version 9 (Graph-Pad Software Inc., San Diego, CA, USA). The colored squares (blue-to-red) represent the relative content of each metabolite (low to high, respectively). The brown, green, purple, blue, and pink characterized metabolites represent the class of sugar and sugar alcohol, phenolic acids, organic acids, amino acids, and fatty acids, respectively. The intermediates are shown in black. Yellow highlighted metabolites have been reported to inhibit melanogenesis.

phenolic acids. The subsequent metabolic activity of the De-E11 starter culture yields a range of compounds, including sugar alcohols, organic acids, amino acids, and fatty acids, with specific metabolites in FUBRS exhibiting inhibitory effects on melanogenesis and/or tyrosinase. Importantly, the valuable insights into the characteristics and metabolite profile of FUBRS derived from this study provide a rational basis for optimizing and controlling fermentation processes. The identified key metabolites can serve as critical biomarkers, enabling the efficient and reproducible production of high-efficacy ingredients

with significant potential in the cosmetic, nutraceutical, and potentially pharmaceutical sectors.

### Funding

This work was supported by the Ratchadapisek Somphot Fund for Postdoctoral Fellowship, Chulalongkorn University and the 90th Anniversary Chulalongkorn University Fund (Ratchadaphiseksomphot Endowment Fund). The funders had no role in study design, data collection and analysis, decision to publish, or preparation of the manuscript.

### Grant Disclosures

The following grant information was disclosed by the authors:
The Ratchadapisek Somphot Fund for Postdoctoral Fellowship, Chulalongkorn University and the 90th Anniversary Chulalongkorn University Fund (Ratchadaphiseksomphot Endowment Fund).

### Competing Interests

The authors declare there are no competing interests.

### Author Contributions

- Orrarat Sangkaew conceived and designed the experiments, performed the experiments, analyzed the data, prepared figures and/or tables, authored or reviewed drafts of the article, and approved the final draft.
- Suttida Kaenboot conceived and designed the experiments, performed the experiments, analyzed the data, prepared figures and/or tables, authored or reviewed drafts of the article, and approved the final draft.
- Thumnoon Nhujak conceived and designed the experiments, authored or reviewed drafts of the article, and approved the final draft.
- Chadin Kulsing performed the experiments, authored or reviewed drafts of the article, and approved the final draft.
- Nuttanee Tungkijanansin performed the experiments, authored or reviewed drafts of the article, and approved the final draft.
- Sittiruk Roytrakul analyzed the data, authored or reviewed drafts of the article, and approved the final draft.
- Chulee Yompakdee conceived and designed the experiments, authored or reviewed drafts of the article, and approved the final draft.

### Data Availability

The raw data is available in the Supplemental Files.

### Supplemental Information

Supplemental information for this article can be found online at http://dx.doi.org/10.7717/peerj.19533#supplemental-information.

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
