# Peer review of "Untargeted metabolomic analyses of fermented unpolished black rice with melanogenesis inhibition activity"

_PeerJ, doi:10.7717/peerj.19533_

## Round 0.1 · original submission · Major Revisions

Dear authors, I ask you to carefully study the reviewers' fundamental comments and improve the manuscript. I hope that the new version of this article will be approved by the reviewers.

Reviewer 1 ·

Basic reporting

It has a clear language and sufficient literature references. However, this paper did not clarify the application of this finding, it has been compared between the polish and un polish black rice. It is obvious to find these difference in the contain of the chemical compounds and their concentrations after fermentation process.
I would recommend to link this finding with its application in the abstract, literature review and in the conclusion.

Experimental design

It matches the journal aims and scope.
Research question is not will define, and needs to clarify the application of this work, and how benefit the scientist community will from this finding. Knowledge gap was not clear in the paper and what is the next step of this work.
Experimental design was good however. What is the point of using 4 different micro organisms together?
It would be better to exam each one first then make mix with two of them then three in different formulations to find which one gives best results.

Validity of the findings

It depends on the aim of this work which is not clear.
Conclusion needs to clarify the benefit and the application of this results.

Additional comments

No additional commenrts

·

Basic reporting

The authors have explained metabolomic analyses of fermented unpolished black rice. However, no significance background information on melanogenesis in black is mentioned in the introduction. Authors should give substantial literature for the same and should also justify the relevance of this in the given study.

Experimental design

Why quantitation is not performed on GC-MS data? What is the relevance of using HPLC separately for quantitative estimation of metabolites?

Validity of the findings

No comments

Additional comments

No comments

Reviewer 3 ·

Basic reporting

The article, namely Untargeted metabolomic analyses of fermented unpolished black rice with melanogenesis inhibition activity, is attractive to metabolomics’ scientists. However, some points are confused and addressed before publication. Minor grammar mistakes are found throughout the manuscript.
Abstract
- Kindly revise the introduction of the abstract. Initially, the authors ought to elucidate the concept of FUBRS (local designation) and subsequently delineate its advantages.
- What is De-E11?
- Omit significant numbers for the results.
- Endeavor to elucidate the correlation among OPLS-DA (Fig. 4), the volcano plot (Fig. 7), and HCA (Figs. 5 and 6) to clarify metabolic pathways.
Introduction
- Paragraph 3, please write local name for fermented rice sap.
- Kindly provide information regarding the definition of metabolomics and the scientific instruments utilized for metabolomic analysis.

Experimental design

-

Validity of the findings

Results and discussion
- Table 2, provide the whole names of TPC, TFC, and TAC beneath the table.
- Lines 296-297: Include additional references, as only two are now cited.
- Kindly identify the chemicals that suppress tyrosinase activity, use the data from Figures 5 and 6 to elucidate this phenomena.
- Why does Fig. 3 display the kojic acid positive control whereas it is absent in Fig. 2?
- Section of melanogenesis inhibition activity, please elucidate why the results of mushroom tyrosinase inhibition activity are not correlated with the melanogenesis inhibition activity.
- Line 332, change into 9 amino acids.
- Figures 5 and 6, what do FUBRS1, 2, and 3, as well as Un-FR1, 2, and 3, represent? What accounts for the discrepancies in treatment patterns across all figures and tables?
- Revise the sentence found in lines 337-340. Determine the names of 27 and 9 metabolites. The data from Fig. 7 are integrated into Figs. 5 and 6.

Additional comments

-

Annotated reviews are not available for download in order to protect the identity of reviewers who chose to remain anonymous.

---

## Round 0.2 · accepted · Accept

Dear Dr. Sangkaew and Dr. Yompakdee, I am pleased to inform you that your article has been accepted for publication in our journal. I hope that you will continue to publish such high-quality articles on this topic in the future.

Reviewer 3 ·

Basic reporting

-

Experimental design

-

Validity of the findings

-

Additional comments

It should be approved since the majority of the addressing has been amended.